# Suppression of Oxidative Stress and Proinflammatory Cytokines Is a Potential Therapeutic Action of *Ficus lepicarpa* B. (Moraceae) against Carbon Tetrachloride (CCl_4_)-Induced Hepatotoxicity in Rats

**DOI:** 10.3390/molecules27082593

**Published:** 2022-04-18

**Authors:** Senty Vun-Sang, Kenneth Francis Rodrigues, Urban J. A. Dsouza, Mohammad Iqbal

**Affiliations:** 1Biotechnology Research Institute, Universiti Malaysia Sabah, Jalan UMS, Kota Kinabalu 88400, Sabah, Malaysia; sentyvunsang2707@gmail.com (S.V.-S.); kennethr@ums.edu.my (K.F.R.); 2Father Muller College of Allied Health Sciences, Father Muller Medical College, Father Muller Road, Kankanady, Mangalore 575002, Karnataka, India; urbandsouza@fathermuller.in

**Keywords:** *F. lepicarpa*, carbon tetrachloride, oxidative stress, proinflammatory cytokines

## Abstract

Local tribes use the leaves of *Ficus lepicarpa* B. (Moraceae), a traditional Malaysian medicine, as a vegetable dish, a tonic, and to treat ailments including fever, jaundice and ringworm. The purpose of this study was to look into the possible therapeutic effects of *F. lepicarpa* leaf extract against carbon tetrachloride (CCl_4_)-induced liver damage in rats. The DPPH test was used to measure the antioxidant activity of plants. Gas chromatography-mass spectrometry was used for the phytochemical analysis (GCMS). Six groups of male Sprague-Dawley rats were subjected to the following treatment regimens: control group, CCl_4_ alone, *F. lepicarpa* 400 mg/kg alone, CCl_4_ + *F. lepicarpa* 100 mg/kg, CCl_4_ + *F. lepicarpa* 200 mg/kg and CCl_4_ + *F. lepicarpa* 400 mg/kg. The rats were euthanized after two weeks, and biomarkers of liver function and antioxidant enzyme status were assessed. To assess the extent of liver damage and fibrosis, histopathological and immunohistochemical examinations of liver tissue were undertaken. The total phenolic content and the total flavonoid content in methanol extract of *F. lepicarpa* leaves were 58.86 ± 0.04 mg GAE/g and 44.31 ± 0.10 mg CAE/g, respectively. *F. lepicarpa*’s inhibitory concentration (IC_50_) for free radical scavenging activity was reported to be 3.73 mg/mL. In a dose-related manner, *F. lepicarpa* was effective in preventing an increase in serum ALT, serum AST and liver MDA. Histopathological alterations revealed that *F. lepicarpa* protects against the oxidative stress caused by CCl_4_. The immunohistochemistry results showed that proinflammatory cytokines (tumour necrosis factor-α, interleukin-6, prostaglandin E_2_) were suppressed. The antioxidative, anti-inflammatory, and free-radical scavenging activities of *F. lepicarpa* can be related to its hepatoprotective benefits.

## 1. Introduction

The liver is the largest organ in the body; it accounts for around 2% of adult body weight, is a dark reddish-brown in colour, is shaped like a cone and weighs roughly 1.5 kg. The liver is found on top of the stomach, right kidney, and intestines in the upper right-hand region of the abdominal cavity, beneath the diaphragm [1,2]. It has an incredible number of functions that support the function of other organs and have an impact on all physiologic systems [3]. Every year, approximately 2 million people die from liver disease around the world [4]. The Global Health Estimates 2019–2020 by the World Health Organization (WHO) noted liver diseases as the 11th most common cause of death worldwide; liver diseases were ranked the 9th most common cause of death in 2019, increasing to the 8th most common cause in the year 2020 in South East Asia (SEA). In Malaysia, it was the 7th leading cause of death over the years 2010–2019 [5,6].

Alcohol-induced cirrhosis, viral-induced chronic liver diseases, environmental toxins, parasitic diseases, hepatitis B and C viruses and hepatotoxic drugs (certain antibiotics, chemotherapeutic agents, high doses of paracetamol, carbon tetrachloride etc.) have been reported to cause liver disease [7,8]. Furthermore, modern liver disease therapies (interferon, colchicine, penicillamine and corticosteroids), which are widely used in the treatment of various diseases, have been found to be ineffective, expensive, and have a significant risk of developing adverse side effects and consequences. This represents a substantial burden of morbidity, especially for government hospitals in treating people diagnosed with liver diseases [9,10]. Despite significant advances in modern medicine, effective medications that boost hepatic function, provide total organ protection, or aid in the regeneration of hepatic cells are not available in the allopathic system. As a result, pharmaceutical alternatives for the treatment of liver disease must be identified, with the goal of making these options more effective and less hazardous. As a response, many plant-based folk medicines are being investigated for their possible antioxidant, phytochemical, and hepato-protective qualities [11].

Keeping this in perspective, the present study was a well-established approach to investigate the herbal plant *Ficus lepicarpa*, which is known to be used as a medicinal herb by Malaysian ethnic groups, utilizing a well-established model of oxidative tissue injury in rats using carbon tetrachloride (CCl_4_) [12]. *F. lepicarpa* is a small tree that belongs to the family Moraceae found in the South East Asia region (Myanmar, Thailand, Malaysia, Indonesia and Philippines, but absent in Singapore). It grows 5–15 m tall and is often found in humid forests, typically on rocky banks of rivers, up to 1700 m altitude [13,14]. *F. lepicarpa* is commonly known as Saraca fig [13,15], and in Malaysia, it is known as ‘Kelupang Gajah’ (Malay) [16]. In Indonesia, it is known as ‘buku-buku’ (Sumatra) or ‘iyubyub etem’ (Javanese), in Thailand, it is known as ‘chalukpho’ (Nakhon Si Thammarat), and in the Philippines, it is known as ‘sulu-talobog’ (Bisaya). The local indigenous tribe in Sabah (North Borneo), Malaysia, locally calls it ‘Ombuwasak’ (Rungus) [17], ‘Tombuwasak’ (Dusun) [18] and ‘Litotobow’ (Murut) [19]. Traditionally, the *F. lepicarpa* leaves are used to treat ringworm [20] and jaundice [18]. Its root is used as a tonic drink and to treat fever [17,18], the ripe fruit is edible, and young leaf shoots are usually eaten as a vegetable or salad [15,18,19]. However, at present, local knowledge of *F. lepicarpa* as a hepatoprotective agent is poorly documented in the scientific literature. Thus far, no one has reported the hepatoprotective effect of *F. lepicarpa*, which is still lacking. Hence, the present study has been undertaken to determine the phytochemical constituents of *F. lepicarpa* along with its potential to protect rats from CCl_4_-induced liver damage and poisoning.

## 2. Results

### 2.1. F. lepicarpa’s In Vitro Antioxidative Activity

Total phenolic content (TPC) concentrations in *F. lepicarpa* methanol extracts were calculated using the equation (y = 4.268x + 0.0436, r^2^ = 0.9939) as gallic acid equivalents (GAE mg/g of extract). TPC in *F. lepicarpa* was found to be 58.86 0.04 mg GAE/g. TFC of *F. lepicarpa* were estimated using the equation (y = 3.35x − 0.019, r^2^ = 0.9937) and expressed as catechin equivalents (CAE mg/g). Total flavonoid content (TFC) obtained was 44.31 ± 0.10 mg CAE/g. The methanol extract antioxidant activity was calculated using its DPPH scavenging activity (a free radical compound). The ability of *F. lepicarpa* extract to scavenge DPPH radicals increased with concentration. At a dosage of 3.73 mg/mL, the methanolic extract of *F. lepicarpa* scavenged DPPH radicals up to 50% (IC_50_).

### 2.2. F. lepicarpa Preliminary Phytochemical Screening 

As indicated in Table 1, preliminary phytochemical screening of *F. lepicarpa* methanol extract revealed the presence of various phytochemicals such as flavonoids, phenols, saponin, steroids, phytosterols, and triterpenoids, whereas alkaloids, tannins, and anthraquinones were undetectable.

### 2.3. F. lepicarpa Gas Chromatography-Mass Spectrometry (GC-MS) Analysis

The chromatogram of GC-MS analysis showing peaks of the number of compounds from the phytochemical constituents of methanol extracts of *F. lepicarpa* is presented in Figure 1. In Table 2, the presence of 30 different bioactive compounds was tabulated along with retention time (Ret. Tim) and area percentage (%). The identified major compounds are 12-Oleanen-3-yl acetate, (3.alpha.)—(21.09%), urs-12-en-24-oic acid, 3-oxo-, methyl ester, (+)- (9.80%) and acetic acid, and 3-hydroxy-6-isopropenyl-4,8a-dimethyl-1,2,3,5,6,7,8,8a-octahydronaphthalen-2-yl ester (13.69%). 

### 2.4. Liver Index and Body Weight

The body weight of the rats was measured every week until they were euthanized. Table 3 shows the body weight, percentage increase in body weight, and liver index of rats from each group. Liver index is the ratio of weight of liver and body weight of the rats expressed as percent of the somatic weight. It is a biomarker that indicates the status of feeding and metabolism. The presence of a large liver signified a high level of metabolic activity, whereas a tiny liver could indicate a shortage of food [21]. The CCl_4_-treated group was found to have a higher liver index than the control group. However, the differences found were not statistically significant. The liver index for the control and plant control groups did not differ significantly. Body weight was reduced as a result of CCl_4_. Animals given *F. lepicarpa* extracts (100, 200, and 400 mg/kg bwt) for two weeks showed a moderate increase in body weight. The proportion of weight gain in the CCl_4_ group was found to be very low when compared to the control and plant control groups.

### 2.5. F. lepicarpa’s Effect on Serum Transaminases: Alanine Aminotransferase (ALT) and Aspartate Aminotransferase (AST)

ALT and AST are the most common liver damage markers detected in the blood. Elevations in these values, which reflect liver function, suggest liver damage. A decrease in these liver enzymes′ activities itself is not considered to be toxicologically significant. Table 4 shows that serum indicators in the CCl_4_ group were substantially higher (*p* < 0.05) as compared to the control and plant control groups. On the marker, the control and plant control groups had identical values and were within the typical laboratory range [22]. Animals given different doses of plant extracts, on the other hand, showed a significant decline (*p* < 0.05) in marker levels before recovering in a dose-dependent manner.

### 2.6. F. lepicarpa’s Effect on Liver Reduced Glutathione (GSH)

Reduced GSH is an intracellular non-enzymatic antioxidant that aids in the defence against free radicals in the liver. Under oxidative stress, GSH levels in tissues are significantly reduced. As shown in Table 5, the level of GSH in the CCl_4_ group decreased by 25% (*p* < 0.05) when compared to the control and plant control groups. When compared to the CCl_4_ treated group, the group pre-treated with *F. lepicarpa* (100, 200, and 400 mg/kg bwt) demonstrated a dose-related increase in GSH, resulting in a recovery of 42, 44, and 59 percent, respectively.

### 2.7. F. lepicarpa’s Effect on Liver Lipid Peroxidation (LPO)

Peroxidation of lipids is a form of oxidative damage to polyunsaturated lipids caused by the reaction of lipids and oxygen, which results in the formation of free radicals and peroxides, which cause cell and organelle damage [23]. The ultimate product of lipid peroxidation is malondialdehyde (MDA). The MDA level in liver homogenate can be examined to assess the degree of liver damage. When compared to the control group, the LPO level in the CCl_4_ treatment group was substantially higher (*p* < 0.05), as shown in Table 5. The control group and plant control group showed similar results. The group pre-treated with *F. lepicarpa* (100, 200 and 400 mg/kg bwt) extract showed lower MDA formation by 85%, 71% and 53%, respectively, in a dose-related manner. Higher dosage treatment group showed a significant decrease (*p* < 0.05) in MDA formation when compared to the CCl_4_ model treatment group. Increased protection against LPO by plant extracts suggests that either the activation or reaction of oxygen and lipids to form free radicals and peroxide enzymes may be inhibited. The chain initiation and propagation, as well as acceleration of chain termination, may have inhibited free radical-mediated lipid peroxidation.

### 2.8. F. lepicarpa’s Effect on Liver Antioxidant Enzymes: Glutathione Peroxidase (GPx), Glutathione Reductase (GR), Glutathione-S-Transferase (GST) and Quinone Reductase (QR)

Enzymatic antioxidants play an essential role in the cell defence system, especially during the healing process. The antioxidant enzymes (GPx and GR) and phase II metabolising enzymes (GST and QR) were studied to learn more about the mechanism of *F. lepicarpa* extract’s protection against CCl_4_ liver damage. Table 6 shows the levels of antioxidant enzymes GPx, GR, GST, and QR in various treatment groups. According to the table, all enzymes in the CCl_4_ model-treated group were significantly depleted (*p* < 0.05) due to increased activity. The plant control group for *F. lepicarpa* (400 mg/kg bwt) had increased antioxidant enzyme activity due to antioxidant supplementation from the plant itself. The antioxidant enzymes were determined to be restored in a dosage-dependent manner in the *F. lepicarpa* -CCl_4_ treated groups; as a result, enzymatic activity increased significantly (*p* < 0.05). Pre-treatment group with *F. lepicarpa* (100, 200 and 400 mg/kg bwt) extracts showed recovery in the antioxidant enzymes GPx (40%, 61% and 63%, respectively), GR (49%, 68% and 85%, respectively), GST (54%, 67% and 75%, respectively) and QR (71%, 88% and 89%, respectively) compared to the CCl_4_ model group.

### 2.9. F. lepicarpa’s Effect on Liver Histopathology

Histopathological changes in liver cells can provide evidence of the biochemical effects of liver protection. As shown in Figure 2A–F, alterations in the histopathology of liver damage were observed, including cell necrosis (N), fatty degeneration (FD), mononuclear cell infiltration (MCI), increase in sinusoidal space (S), cell derangement (CD) of hepatocytes (H) and congestion of the central vein (CV). In the saline control group (Figure 2A) and plant control group (Figure 2F), it was discovered that the central vein, hepatocytes, and sinusoidal space were all normal. In contrast, the CCl_4_ model group (Figure 2B) resulted in severe histopathological alterations, including massive hepatocytes necrosis, steatofibrosis characterized by fatty degeneration or accumulation in hepatocytes, increased mononuclear infiltration, increased sinusoidal space forming fibrous bridges between cells due to formation of liver pseudo lobules and an abnormal asymmetrical location of central veins. However, in the group pre-treated with *F. lepicarpa*-CCl_4_ (Figure 2C–E), these changes were attenuated in a dosage-dependent manner. The inflammation of liver cells and fatty degeneration was significantly reduced, particularly at 400 mg/kg bwt (Figure 2E). The hepatocyte cord showed less disarray, as well as the repair of cellular boundaries.

### 2.10. F. lepicarpa’s Effect on Liver Proinflammatory Cytokines

An inflammatory response is elicited by CCl_4_-induced liver injury. This activates Kupffer cells, which mediate the action of cytokines such as interleukin-6 (IL-6), tumour necrosis factor-alpha (TNF-α) and prostaglandin E_2_ (PGE_2_). Due to this, the effect of *F. lepicarpa* on proinflammatory marker expression for TNF-α, IL-6 and PGE_2_ was also investigated (Figure 3, Figure 4 and Figure 5, respectively). The immuno-staining on the liver section of the saline control (Figure 3A, Figure 4A and Figure 5A) and plant control groups (*F. lepicarpa* (400 mg/kg bwt)) (Figure 3F, Figure 4F and Figure 5F) showed a complete absence of intense brown colouration, which indicates that the markers did not present in the cells; thus, no liver cell damage was detected. However, compared to the control group, the CCl_4_ model group (Figure 3B, Figure 4B and Figure 5B) showed an intense brown colouration, which demonstrates that marker-positive cells were present. In contrast, a decrease in the appearance of the intense brown colouration of these markers was observed in *F. lepicarpa*-CCl_4_ treated group in a dosage-dependant manner (Figure 3C–E, Figure 4C–E and Figure 5C–E).

## 3. Discussion

This investigation set forth to identify the phytochemical contents of *F. lepicarpa* plants, as well as their potential to provide a hepatoprotective effect on induced liver injury using CCl_4_ in Sprague-Dawley rats. These plants were chosen for their therapeutic characteristics, which have been widely utilised as traditional medicine in local communities to treat a range of illnesses, including jaundice [18], ringworm infection [20], and fever [17]. These plant products are also consumed as tonics, vegetables, and edible ripe fruits, in addition to being utilised as alternative medicine. Despite their widespread use as a medicinal and food source, there is still a dearth of scientific evidence about these plants in the literature.

The drying method of fresh leaves of *F. lepicarpa* involved maintaining a temperature of 40 °C to prevent the degradation of the bioactive compounds. The selection of drying methods for *F. lepicarpa* was made due to a previous study that dried the leaves under natural conditions. The leaves turned out to be brown and blackish in colour, and this, in turn, affected the total phenolic and total flavonoid assay as well the antioxidant assay [18]. The selection of the drying method prior to further experimentation is important because it will affect the outcome of the downstream research. Drying is regarded as the most crucial step in the post-harvest process due to its importance in limiting enzymatic degradation and microbial growth while preserving the plant′s beneficial properties [24]. The drying procedure alters the chemical composition of the herbal medical preparation, lowering its quality [25]. Previous research shows that the selection of drying methods on chrysanthemum flower heads shows the advantages of oven drying over sun drying and shade drying in terms of the TPC, TFC and antioxidant properties [26]. The method of extraction and selection of solvent in the extraction are required based on the targeted bioactive compound and the plants type [27,28]. Earlier reports show that flavonoid and phenolic compounds were abundant in methanol extraction, while carotenoids and capsaicinoids were found to be abundant in hexane extraction [29]. This stressed that selectivity is significantly tied to the target compound′s solubility in the solvent itself.

The methanol extract of *F. lepicarpa* leaves contained elevated levels of TPC and TFC as well as the ability to scavenge free radicals as measured by DPPH assay. Plant phenolics and flavonoids are phytochemical substances present in both edible and inedible plant components that have been shown to have a variety of biological functions, in which antioxidant and anti-inflammatory properties are included. Because of their redox properties, phenolic compounds can act as singlet oxygen quenchers, reducing agents and hydrogen donors, which contribute to their scavenging properties [30]. Flavonoids are secondary metabolites having antioxidant activity, the effectiveness of which is determined by the amount and location of free OH groups. The DPPH test is a simple, acceptable, and extensively used method for determining a plant extract’s radical scavenging potency [31]. Similarly, previous research has discovered a strong positive connection between antioxidant activity and total flavonoid and total phenolic concentrations in celery [32].

Plant phytochemical compounds have been reported to participate in a variety of biological activities, such as antibacterial, antifungal, antioxidant, anti-cancer and anti-inflammatory activity [33]. Early phytochemical screening revealed that *F. lepicarpa* contained flavonoids, phenols, saponin, steroids, phytosterols, and triterpenoids. Literature has documented that all the present secondary metabolite compounds have potential health-promoting properties [34].

The methanol extract of *F. lepicarpa* leaves under GC-MS analysis revealed the presence of a diversity of bioactive compounds. The major compounds, which include 12-Oleanen-3-yl acetate, (3.alpha.) and urs-12-en-24-oic acid, 3-oxo-, methyl ester, (+)-, are presumed to be pentacyclic triterpenoid isomers. Both compounds are also known as β-amyrin acetate and α-amyrin acetate, respectively [35]. *Ficus racemosa*, *F. cordata*, *F. palmata*, *F. thumbergii*, *F. sur* and *F. sycomorus* have all been found to contain α-amyrin acetate and β-amyrin acetate [36]. The existence of these compounds in *F. lepicarpa* indicates that it shares the biochemical profile of the genus. Previous reports show that these compounds possess anti-inflammatory properties, which are used by pharmaceuticals to treat wounds, ulcers, and joint, bone and liver infections [37,38].

The body weight of the rats was recorded to keep track of their health. The increase percentage of body weight inferred that all individuals were healthy. The low percentage increase in body weight in CCl_4_ model groups and the group pre-treated with plant extracts when compared to control and plant control groups implies a decrease in food and water intake, which is likely to be the result of CCl_4_ toxicity. CCl_4_ has been shown in previous studies to limit nutritional consumption due to maldigestion or malabsorption caused by gastrointestinal disorders [39]. In a dose-dependent manner, *F. lepicarpa* promotes body weight recovery. Higher doses of plant extract show improved recovery. This backs up a recent study that found that giving walnut extract to rats in CCl_4_ recovered their body weight loss [40]. Liver index in all groups did not differ. A previous study found that lipid and collagen build-up raises the liver index. However, due to acute CCl_4_ exposure in rats, it may not increase to the point that it significantly influences the liver index [41].

The toxicity of CCl_4_ was validated by measuring ALT and AST levels in the blood. The levels of enzymes increase dramatically after the final CCl_4_ therapy. The bioactivation of the cytochrome *P*-450 system as a result of CCl_4_ toxicity forms a toxic reactive trichloromethyl peroxyl radical (CCl*_3_). This radical further attacks membrane lipids to initiate a chain reaction, resulting in the peroxidation of membrane lipids, leading to hepatocellular damage. The production of free radicals (trichloromethyl and peroxytrichloromethyl) due to CCl_4_ results in the leakage of cytoplasmic ALT and AST enzymes into the circulatory system. The high enzymatic levels indicate that the liver structure has been severely damaged [42,43]. With the administration of various doses of *F. lepicarpa* extract, the levels of this enzyme marker were restored in a dose-related manner. The findings of this study are in agreement with previous research in which plant extract restored serum markers in the blood after being induced with CCl_4_ [44]. The stability of the plasma membrane and hepatocyte repair could explain the recovery of these damages. The findings suggest that *F. lepicarpa* methanol extracts may protect the liver from CCl_4_-induced damage.

Antioxidant enzymes have been proven in previous research to be the first line of defence against reactive oxygen species (ROS) and other free radicals [44,45,46,47,48]. Previous studies showed that antioxidants such as taurine, N-acetylcysteine (NAC) and α-tocopherol revealed the recovery of hepatocytes against induced toxicity by medicines such as triazole rizatriptan, thioridazine and citalopram [49,50,51]. The antioxidant activity of *F. lepicarpa* against CCl_4_-induced ROS in rats is investigated in this work. The levels of GSH, LPO, GPx, GR, GST, and QR in hepatic tissues were measured, which reflected the ROS production generated by CCl_4_ in the rats′ liver. ROS generation is also associated with a decrease in the mitochondrial membrane potential followed by DNA fragmentation and an increase in the expression of pro-apoptotic and inflammatory markers. The administration of *F. lepicarpa* extracts in rats showed recovery in the levels of GSH, LPO, GPx, GR, GST, and QR in CCl_4_-induced toxicity which, in general, reflects the recovery of mitochondrial membrane permeability and, thus, decreases in the ROS formation.

Reduced glutathione (GSH) is a low molecular weight non-enzymatic antioxidant that is normally present in all cell types. It serves as a first line of defence against free radicals and functions as a co-substrate for other antioxidants. Due to CCl_4_ toxicity, free radicals are produced, causing oxidative stress, a decline in mitochondrial membrane potential, inflammatory cell death, and tissue damage as a result of membrane lipid disruption. GSH levels are often reduced upon elevation of oxidative stress. The decrease in GSH levels could be related to an increase in cell usage to scavenge free radicals formation produced by CCl_4_ [52]. In the current investigation, CCl_4_-administered rats’ liver GSH levels were considerably lower than in the control groups, which is consistent with earlier research [45,46]. The toxicity of CCl_4_ is reduced by pretreatment of *F. lepicarpa*. The mechanism of *F. lepicarpa* liver protection against CCl_4_ toxicity may include the restoration of GSH levels. The possible mechanism of the hepatoprotective role of *F. lepicarpa* might be due to the presence of bioactive compounds which neutralize reactive oxidants directly, enhance the endogenous antioxidant defence system, and increase the steady-state GSH and/or the synthesis rate of GSH to enhance the protection against oxidative stress and restore the mitochondrial membrane potential caused by CCl_4_’s toxicity effect [52].

Lipid peroxidation is the cause of cell membrane damage and the genesis of liver injury caused by free radical offshoots of CCl_4_. Free radicals primarily attack the phospholipid bilayers of cellular and subcellular membranes. Increased levels of MDA (the end product of lipid peroxidation) indicate increased lipid peroxidation [53]. In this study, the MDA level increased considerably in the CCl_4_-treated model group. When animals were pre-treated with a *F. lepicarpa* extract, the amount of MDA in their livers was reduced moderately in a dose-related manner. This finding is in good agreement with previous studies, where CCl_4_-induced treatment elevated the MDA level [44,45,46,47,48]. The reduction in MDA levels indicated that lipid peroxidation was being inhibited alongside an increase in antioxidative defence mechanisms to prevent free radical synthesis and development, which would cause oxidative damage [54].

In addition to non-enzymatic antioxidants (GSH and LPO), enzymatic antioxidants (GPx and GR) and phase II metabolizing enzymes (GST and QR) also play an important role in the defence mechanism against the free radicals. These antioxidant enzymes play a crucial role in the detoxification process and provide protection during the healing process by scavenging free radicals during oxidative stress. However, these antioxidant enzymes are susceptible to oxidation [55]. The activities of these enzymes were significantly reduced in the CCl_4_ model treatment group when compared to the normal and control plant group. The findings in this research are in accordance with previously published research [44,45,46,47,48]. The addition of methanol extract of *F. lepicarpa* leaves to rats’ intake increased enzyme activity, demonstrating the plant′s antioxidant and hepatoprotective properties.

Histopathological investigations are required to support biochemical findings [56]. Histopathological observation in the CCl_4_ treatment model indicated that CCl_4_ induced fibrosis, cirrhosis, and hepatocarcinoma. *F. lepicarpa* reduced severe liver injury caused by CCl_4_ in a dose-dependent manner. The histological manifestations confirmed the biochemical findings and clearly suggested that *F. lepicarpa* methanol extract has a significant hepatoprotective effect against CCl_4_-induced oxidative stress. The histological manifestations in this study are consistent with the findings of other authors [44,45,46,47,48].

In CCl_4_-treated rats, the inhibitory effects of the *F. lepicarpa* methanol extract on the overexpression of proinflammatory cytokines TNF-α, IL-6, and PGE_2_ were also investigated. The proinflammatory markers of TNF-α, IL-6, and PGE_2_ are strictly under the regulation of nuclear factor (NF)-kappa β transcription, and they promote inflammation of cells, vascular permeability, and proliferation. When the NF-κβ synthesis pathway is activated, it generates an inflammatory response that leads to cellular death. Unless the NF-κβ synthesis pathway is blocked by an exogenous antioxidant, this scenario produces severe liver damage [57,58]. TNF-α, IL-6, and PGE_2_ overexpression in rats treated with CCl_4_ was significantly reduced in rats treated with *F. lepicarpa* extract. These immune-modulatory effects may be owing to the presence of important bioactive substances such as 12-Oleanen-3-yl acetate, (3.alpha.)-, acetic acid, 3-hydroxy-6-isopropenyl-4, 8a-dimethyl-1,2,3,5,6,7,8, 8a-octahydronaphthalen-2-yl ester and urs-12-en-24-oic acid, 3-oxo-, methyl ester, (+)-. The proinflammatory cytokines manifestations in this study are consistent with findings of other research [45,47,48].

## 4. Materials and Methods

### 4.1. Chemicals

Carbon tetrachloride (CCl_4_) and other chemicals used in this study were obtained from Sigma Aldrich (St. Louis, MO, USA). Solvents of analytical grade or GC grade were from Fisher Scientific (Hampton, NH, USA). Alcohol, acid alcohol, blue buffer, eosin, haematoxylin, xylene and DPX mounting medium for histological assessment were purchased from Leica Biosystem (Wetzlar, Germany). Dako provided the immunohistochemistry antibodies and reagents (Glostrup, Denmark).

### 4.2. Sample Collection and Preparation

Fresh *Ficus lepicarpa* leaves (voucher number: SVS 001) were collected from Kg. Morion, Tandek, Kota Marudu, Sabah, Malaysia, in the month of November 2017. Species of the *Ficus* were authenticated by a botanist from the Institute for Tropical Biology and Conservation (IBTP), Universiti Malaysia Sabah. The oven-dried leaves (40 °C) were ground using a blender and macerated in methanol at the ratio of 1:10 (*w*/*v*) for 72 h at room temperature. The extract was filtered through Whatman filter paper No.1, and using a vacuum rotary evaporator, methanol residues were extracted from the extract. The samples were stored at −80 °C for 24 h before being lyophilized with a freeze drier. The freeze-dried samples were then kept in the freezer for subsequent examination [59].

### 4.3. Total Phenolic Content (TPC)

The *F. lepicarpa* methanol leaf extract’s total phenolic content was estimated spectrophotometrically at 725 nm according to the Folin–Ciocalteu method with slight modification [47,60]. The blank had the same constituents except that the extract was replaced by distilled water. All the analysis was repeated three times, and the mean value of absorbance was obtained. Gallic acid dilutions (0.1–0.5 mg/mL) were used as the standard for preparing the calibration curve. The total phenolic content was expressed as mg gallic acid equivalents (GAE) per gram of extract.

### 4.4. Total Flavonoid Content (TFC)

The flavonoid content of *F. lepicarpa* extract was estimated using the aluminium chloride (AlCl_3_) colorimetric method at 510 nm with slight modification [48,61]. In this assay, catechin dilutions (0.01–0.1 mg/mL) were used as a standard to make the calibration curve. The total flavonoids in the extracts were calculated in triplicate, and the results were averaged. The total flavonoid content of the extract was expressed in mg of catechin equivalents (CAE) per gram.

### 4.5. Radical Scavenging Assay (DPPH)

The DPPH assay was used to estimate the free radical scavenging activity of *F. lepicarpa* leaf methanol extract [62]. DPPH reduction was calorimetrically measured at 517 nm against a blank after 1 h incubation in the dark with DPPH. The experiments were carried out in triplicate. The percentage inhibition was calculated using the following formula:Radical Scavenging Activity (RSA) (%) = [(A_control_ − A_sample_)/A_control_] × 100

Here, A_control_ is the absorbance of the control (solution without extract or standard), and A_sample_ is the absorbance in the presence of the extract or standard of various concentrations. The graph of the percentage of RSA versus extract concentration was used to calculate the 50% of extract inhibition concentration (IC_50_). The slope of the linear regression was used to calculate the values.

### 4.6. Phytochemical Screening

The phytochemical screenings of the crude extracts were carried out using a published standard method. The methods were used to identify the following phytochemicals in the extract: alkaloids (Wagner’s test), steroids (Liebermann–Burchard test) [63], flavonoids (alkaline reagent test) [64], tannins (Braymer’s test), triterpenoids (Salkowki’s test) [65], saponin (foam test) [66], phenols (ferric chloride test), phytosterols [67] and anthraquinones [68]. All screening was done in triplicate.

### 4.7. Gas Chromatography-Mass Spectrometry Analysis (GC-MS)

GC-MS analysis was carried out using an Agilent 7890A gas chromatograph coupled with an Agilent 5975C mass spectrometer inert XL EI/Cl MSD with an ionization energy of 70 eV and a capillary column HP-5MS (30 m × 0.25 mm × 0.25 μm). The injection volume was adjusted to 1 µL in the splitless mode. The injector temperature was set at 250 °C using pure helium gas as a carrier at a constant flow rate of 1.0 mL/min. Separation of metabolites was performed at a temperature program from 100 °C, which was held for 3 min and then gradually increased from 100 °C to 180 °C with steps of 15 °C/min and from 180 °C to 300 °C with steps of 5 °C/min. It was then held at 300 °C for 10 min. Identification of the chemical compounds of the extract was based on gas chromatography retention time and the concentration of the compounds in the chromatogram, with a computer matching the mass spectra with those of standards from the National Institute of Standards and Technology (NIST) library. Along with a blank solvent, each analysis was performed in triplicate.

### 4.8. Animal Experiments

Adult Sprague-Dawley male rats weighing 200 to 250 g were obtained from Biotechnology Research Institute’s Animal Breeding House located at the Animal Biosafety Lab (ABSL) facilities of Universiti Malaysia Sabah. During the study, the animals were raised in a laboratory animal house using dried corn bedding in plastic (polypropylene) cages with a controlled environment (25 ± 3 °C and 50% humidity) and free access to a continuous food and water supply. The protocols for animal use were in accordance with the guidelines of care and use of laboratory animals by the National Academy of Sciences [69] and approved by the Animal Ethics Committee of Universiti Malaysia Sabah (UMS/PPPI1.3.2/800-2/1/17 Jilid 4 [3]). The animals were acclimatized for a week in standard laboratory environmental conditions and randomly assigned to control and experimental groups. Thirty-six adult male rats were randomly assigned to six groups of six rats each and treated as follows:Group 1: Normal control (were not given any treatment).Group 2: CCl_4_ (1.0 mL/kg bwt).Group 3: CCl_4_ + *F. lepicarpa* (100 mg/kg bwt).Group 4: CCl_4_ + *F. lepicarpa* (200 mg/kg bwt).Group 5: CCl_4_ + *F. lepicarpa* (400 mg/kg bwt).Group 6: *F. lepicarpa* (400 mg/kg bwt) (plant control alone).

The CCl_4_ was given orally at a dose of 1.0 mL/kg bwt in corn oil (1:1). A distilled water extract suspension was prepared, and different doses of *F. lepicarpa* extract (100, 200, and 400 mg/kg b wt.) were administered orally to the animals via gastric gavage needle for 14 days, followed by two doses of CCl_4_ on the 13^th^ and 14^th^ days. The selection of doses of plant extracts for the in vivo experiment and the duration of the treatment period were based on our own preliminary studies. All of these rats were decapitated within 2 h of the last CCl_4_ dose. After cervical dislocation, blood was taken from the posterior vena cava before the heart stopped beating. The clotted blood was centrifuged for 30 min at 2000× *g*, and the serum was kept at −80 °C for serum transaminase (ALT and AST) assays. The extracted liver tissues were perfused using chilled 0.85% *w*/*v* NaCl to remove from extraneous materials and to obtain a homogenate or post-mitochondrial supernatant (PMS). A small portion of liver tissue was homogenized in cooled phosphate buffer (0.1 M, pH 7.4), which was then frozen at −80 °C for subsequent biochemical analysis and antioxidant enzyme assays. To quantify the activity of the quinone oxidoreductase assay, a part of the homogenate was ultracentrifuged at 105,000× *g* for 1 h at 4 °C to yield cytosolic components. The residual liver tissues were fixed in 10% natural formalin buffer for histopathological and immunohistochemistry analysis. The following equation was used to calculate liver indices: the liver index is calculated as the sum of the liver wet weight (g) and the final body weight (g) multiplied by 100.

### 4.9. Serum Transaminases (AST and ALT)

The colorimetric method developed by Reitman and Frankel was used to measure serum alanine aminotransferase (ALT) and serum aspartate aminotransferase (AST) [22,70], which uses buffered enzyme substrate (L-alanine (ALT) or L-aspartate (AST) and α-ketoglutarate) dinitrophyenylhydrazine (DNPH) as a colouring reagent which reacts with pyruvate standard (2 mM) and forms a brown-coloured hydrazone complex in the alkaline conditions. The developed colour was directly proportional to enzyme activities and colour intensities, which were measured at 510 nm. Sodium pyruvates of different concentrations were used to plot a standard graph, from which the unknown serum transaminases were assessed.

### 4.10. Reduced Glutathione (GSH)

The reduced GSH content was determined by the previously established method [71]. The reaction mixture consisted of liver homogenates (10% *w*/*v*), phosphate buffer (0.1 M, pH 7.4), sulfosalicylic acid (4% *w*/*v*) and 5,5-dithiobis-2-nitrobenzoic acid (4 mg/mL) in a total volume of 3.0 mL in which a yellow colour formed. It was immediately read at 412 nm on a visible spectrophotometer. Results are given as micromoles of reduced glutathione per gram of tissue calculated with a molar extinction coefficient of 1.36 × 10^3^ M^−1^ cm^−1^.

### 4.11. Lipid Peroxidation (TBARS Content)

Lipid peroxidation was assessed by measuring thiobarbituric acid reactants (TBARs) productions following a previously described method [72,73]. The reaction mixture consisted of liver homogenates (10% *w*/*v*) and trichloroacetic acid (TCA) (10% *w*/*v*). The supernatant of the mixture after centrifuging with thiobarbituric acid (TBA) (0.67% *w*/*v*) in a total volume of 2.0 mL was read at 535 nm. The results were represented as nanomoles of MDA produced per gram of tissue computed using a molar extinction coefficient of 1.56 × 10^5^ M^–1^ cm^–1^.

### 4.12. Glutathione Peroxidase (GPx)

The presence of glutathione peroxidase (GPx) converts glutathione (GSH) to glutathione disulphide (GSSH) with the reaction of hydrogen peroxide (H_2_O_2_). This was quantified according to a method described earlier [74,75]. The enzyme activity was calculated using a molar extinction coefficient of 6.22 × 10^3^ M^−1^ cm^−1^ and reported as nanomoles of NADPH oxidized every minute for each milligram of protein. The reaction mixture consisted of liver homogenates (10% *w*/*v*), GPx working solution (phosphate buffer (0.1 M, pH7.4), ethylenediaminetetraacetic acid (EDTA) (0.05 mM)), hydrogen peroxide (H_2_O_2_) (0.019 mM), β-nicotinamide adenine dinucleotide phosphate reduced (NADPH) (0.1 mM), sodium azide (NaN3) (1 mM) and reduced glutathione (GSH) (1 mM) in a total volume of 2.335 mL. The reading was recorded every 30 s for 3 min at 340 nm.

### 4.13. Glutathione Reductase (GR)

The glutathione reductase (GR) enzyme transforms oxidised glutathione (GSSG) to reduced glutathione (GSH) and NADPH to NADP^+^ at the same time. This was determined based on a previously described method [75,76]. The enzyme activity was delineated as nanomoles of NADPH oxidized per minute per milligram of protein by a molar extinction coefficient of 6.22 × 10^3^ M^−1^ cm^−1^. The reaction mixture reading was recorded at seven intervals every 30 s for 3 min at 340 nm. The mixture of a total volume of 2.45 mL consists of liver homogenates (10% *w*/*v*), GR working solution (phosphate buffer (0.1 M, pH 7.4) and EDTA (0.5 mM)), oxidized glutathione (1 mM) and NADPH (0.1 mM).

### 4.14. Glutathione S-Transferase (GST)

The demonstration of liver homogenate glutathione S-transferase (GST) initializes the reduced glutathione (GSH) to react with 1-chloro 2,4 dinitrobenzene (CDNB). An ultraviolet chromogenic substrate used to form a CDNB conjugate (CDNB-SG) was employed using a previously described method [77,78]. The changes in absorbance were recorded every 30 s for 3 min at a wavelength of 340 nm and calculated using a molar extinction coefficient of 9.6 × 10^3^ M^−1^ cm^−1^ as nanomoles of CDNB conjugate formed per minute per miligram of protein. The 3.0 mL total volume reaction fusion consisted of homogenates (10% *w*/*v*), CDNB (1 mM), phosphate buffer (0.1 M, pH 7.4) and reduced glutathione (1 mM).

### 4.15. Quinone Reductase (QR)

In the presence of quinone, the reduction of dichlorophenolindophenol (DCPIP) as a two-electron acceptor redox dye is catalysed, and NADPH is oxidised to NADP^+^ in liver homogenate. This was estimated following previously published methods [79,80]. The enzyme reaction was quantified at a wavelength of 600 nm based on the disappearance of DCPIP with 30 s intervals for 3 min. The enzyme reaction was evaluated using a molar extinction coefficient of 2.1 × 10^4^ M^−1^cm^−1^ as nanomoles of DCPIP reduced every minute for each milligram of protein. The test mixture was made up of the following components: cytosolic fraction (10% *w*/*v*), QR working solution (Tris-HCl buffer (0.025 M, pH 7.4), bovine serum albumin (BSA) (1 mg/mL)), 2,6-dichlorophenolindophenol (DCPIP) (2.4 mM), NADPH (0.1 mM), flavin adenine dinucleotide (FAD) (150 μM) and Tween 20 (1% *w*/*v*).

### 4.16. Histopathological Assessment

The liver tissue was fixed in 10% natural buffered formalin. It was then cut into approximately 50 mm thick sections, dehydrated with a series of increasing (70–100%) alcohol concentrations and three series of clearing agents using xylene solvent and perfused with three series of paraffin wax liquid before being embedded in solid paraffin blocks. The tissues blocks were cut into thin ribbon sections (4–5 µm) and stained with haematoxylin and eosin (H&E). A pathologist who was unaware of the sample assignment of experimental groups examined the liver tissue segments using a high-resolution light microscope with photographic services.

### 4.17. Immunohistochemical Assessment

For immunohistochemistry, the fixed liver tissues in 10% natural buffered formalin were processed according to the histological method of tissue processing until being embedded in solid paraffin blocks. The paraffin-embedded tissue was cut at 4 µm, mounted on positively charged slides and dried at 58 °C for 60 min. The slides were then deparaffinised and rehydrated through two series of xylene and a degraded series of alcohol (two series of 100%, one series each of 95%, 70% and 50%) before being rinsed with deionized water and rehydrated with a wash buffer. After microwave antigen retrieval using sodium citrate buffer, the slides were then incubated in 3% hydrogen peroxide to block endogenous peroxidase before being incubated with primary antibodies (rabbit polyclonal antibodies) of interleukin-6 (IL-6) (1:500), prostaglandin E_2_ (PGE_2_) (1:500) and tumour necrosis factor-alpha (TNF-α) (1:500). Signal enhancement was performed using horseradish peroxidase (HRP), and antibody binding was detected by the 3, 3′-Diaminobenzidine (DAB) chromogenic substrate (brown colouration) based on the desired level of colour intensity. The slides were then counterstained with Harris haematoxylin, dehydrated, and mounted with xylene (DPx). Immunoreactivity was measured by a pathologist who was not informed of the sample experimental groups. The appearance of an intense brown colouration on liver sections demonstrated the presence of these markers on the cells.

### 4.18. Statistical Analysis

The findings of this investigation were expressed as mean ± standard error (mean ± SE). To determine the significance of the differences between the control and experimental groups, statistical analysis (SPSS statistical analysis software) was performed using one-way analysis of variance (ANOVA) for repeated measurements with the significance level set at *p* < 0.05.

## 5. Conclusions

In conclusion, the findings of this study suggest that significantly decreased levels of hepatic GSH and lipid peroxidation (MDA) along with normalizing activities of antioxidant enzymes and serum markers (ALT, AST) suggest that *F. lepicarpa* extract has protective effects against CCl_4_-induced hepatotoxicity by reducing the oxidative stress. A reduction in the histopathological alteration and proinflammatory cytokine expression in the liver further supports the biochemical findings. We conclude that *F. lepicarpa* extract could be used as a therapeutic agent to protect the liver from oxidative damage. Prior to considering *F. lepicarpa* as a therapeutic agent, the mechanism of its action, pharmokinetic investigations as well as bioavailabity of its bioactive constituents are essential and needed.

## Figures and Tables

**Figure 1 molecules-27-02593-f001:**
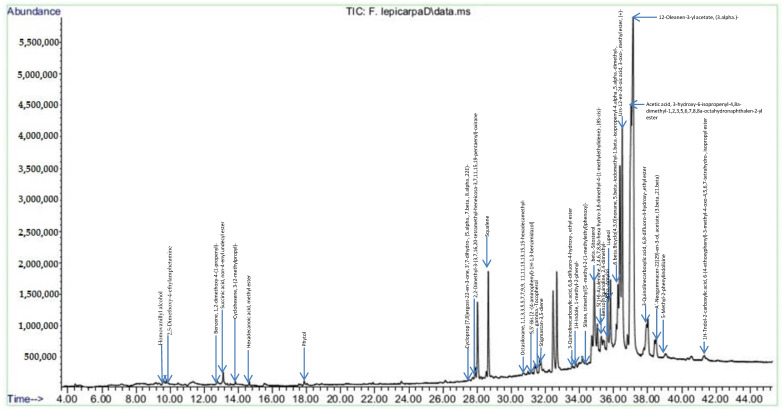
Chromatogram of GC-MS analysis of *F. lepicarpa*.

**Figure 2 molecules-27-02593-f002:**
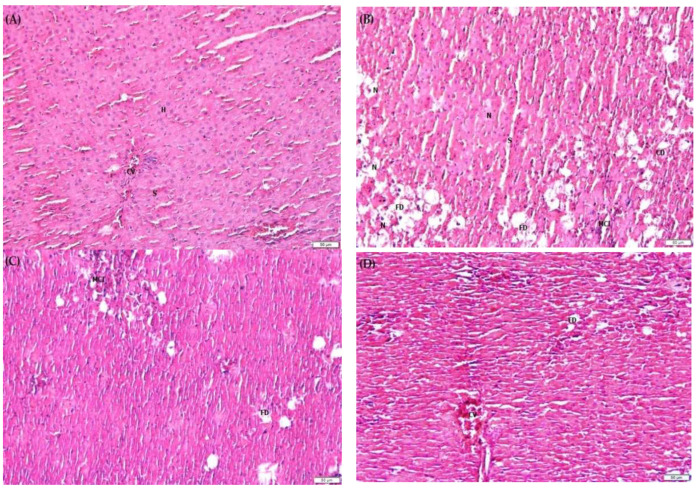
Photomicrographs of rat liver sections stained with haematoxylin and eosin from each group. Total magnifications 10×. (**A**) Control group, normal histology. (**B**) CCl_4_ (1.0 mL/kg bwt)-induced necrosis, fatty acid degeneration, derangement of hepatocytes. (**C**) *F. lepicarpa* (100 mg/kg bwt + CCl_4_); slight repairing of hepatocytes. (**D**) *F. lepicarpa* (200 mg/kg bwt + CCl_4_) repairing of hepatocytes. (**E**) *F. lepicarpa* (400 mg/kg bwt + CCl_4_); repairing of hepatocytes. (**F**) *F. lepicarpa* (400 mg/kg bwt) (normal histology).

**Figure 3 molecules-27-02593-f003:**
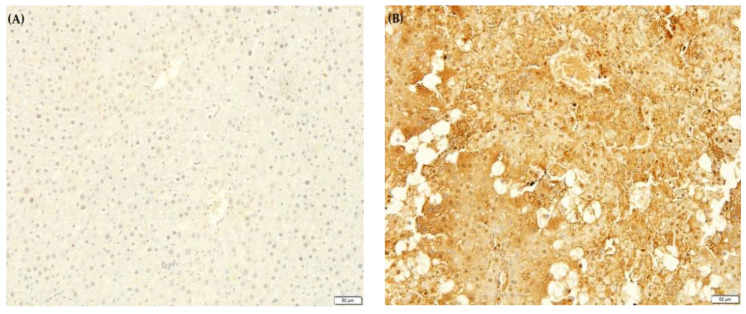
The effect of *F. lepicarpa* extract on the proinflammatory marker tumour necrosis factor-alpha (TNF-α). Total magnifications 10×. (**A**) Control group, normal. (**B**) CCl_4_ (1.0 mL/kg bwt), high expression of TNF-α. (**C**) *F. lepicarpa* (100 mg/kg bwt + CCl_4_), low expression of TNF-α. (**D**) *F. lepicarpa* (200 mg/kg bwt + CCl_4_), low expression of TNF-α. (**E**) *F. lepicarpa* (400 mg/kg bwt + CCl_4_), low expression of TNF-α. (**F**) *F. lepicarpa* (400 mg/kg bwt, plant control), normal.

**Figure 4 molecules-27-02593-f004:**
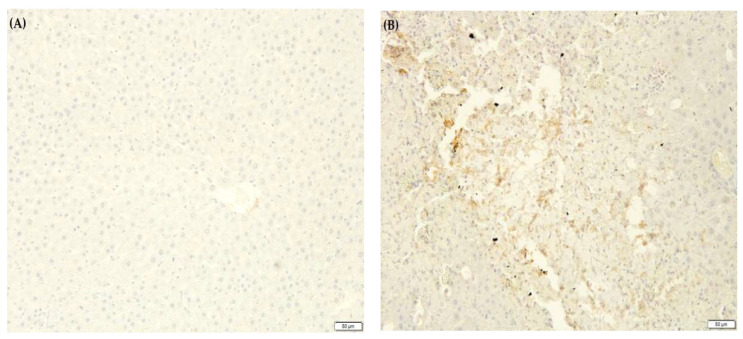
The effect of *F. lepicarpa* extract on the proinflammatory marker interleukin 6 (IL-6). Total magnifications 10×. (**A**) Control group, normal. (**B**) CCl_4_ (1.0 mL/kg bwt), high expression of IL-6. (**C**) *F. lepicarpa* (100 mg/kg bwt + CCl_4_), low expression of IL-6. (**D**) *F. lepicarpa* (200 mg/kg bwt + CCl_4_), low expression of IL-6. (**E**) *F. lepicarpa* (400 mg/kg bwt + CCl_4_), low expression of IL-6. (**F**) *F. lepicarpa* (400 mg/kg bwt, plant control), normal.

**Figure 5 molecules-27-02593-f005:**
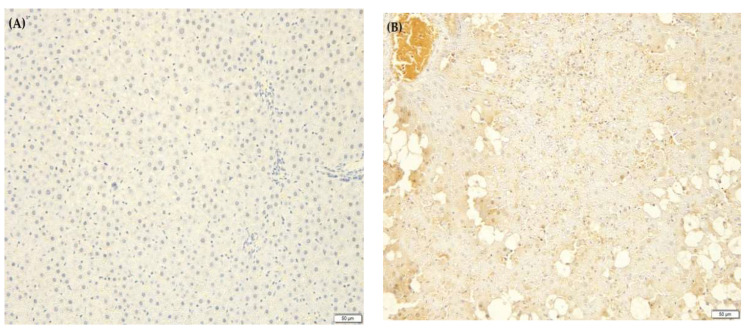
The effect of *F. lepicarpa* extract on the proinflammatory marker prostaglandin E_2_ (PGE_2_). Total magnifications 10×. (**A**) Control group, normal. (**B**) CCl_4_ (1.0 mL/kg bwt), high expression of PGE_2_. (**C**) *F. lepicarpa* (100 mg/kg bwt + CCl_4_), low expression of PGE_2_. (**D**) *F. lepicarpa* (200 mg/kg bwt + CCl_4_), low expression of PGE_2_. (**E**) *F. lepicarpa* (400 mg/kg bwt + CCl_4_), low expression of PGE_2_. (**F**) *F. lepicarpa* (400 mg/kg bwt, plant control), normal.

**Table 1 molecules-27-02593-t001:** Phytochemical constituents of leaf extract of *F. lepicarpa*.

Phytochemical Constituents	Presence (+) or Absence (−)
Alkaloids (Wagner’s test)	−
Flavonoids (alkaline reagent test)	+
Tannins (Braymer’s test)	−
Saponin (foam test)	+
Phenols (ferric chloride test)	+
Steroids (Liebermann–Burchard test)	+
Anthraquinones	−
Phytosterols	+
Triterpenoids (Salkowki’s test)	+

**Table 2 molecules-27-02593-t002:** Phytochemical contents of *F. lepicarpa* identified using GCMS.

No.	Ret. Time (min)	Compound	Area (%)
1.	9.595	Homovanillyl alcohol	0.24
2.	9.773	2,5-Dimethoxy-4-ethylamphetamine	0.16
3.	12.756	Benzene, 1,2-dimethoxy-4-(1-propenyl)-	0.16
4.	13.111	Succinic acid, non-4-enyl undecyl ester	0.42
5.	13.832	Cyclohexene, 3-(2-methylpropyl)-	0.07
6.	14.637	Hexadecanoic acid, methyl ester	0.11
7.	17.865	Phytol	0.13
8.	27.762	Cycloprop [7,8]ergost-22-en-3-one, 3′,7-dihydro-, (5.alpha., 7.beta., 8.alpha.,22E)-	0.13
9.	27.880	2,2-Dimethyl-3-(3,7,16,20-tetramethyl-heneicosa-3,7,11,15,19-pentaenyl)-oxirane	0.44
10.	28.660	Squalene	2.85
11.	30.727	Octasiloxane, 1,1,3,3,5,5,7,7,9,9, 11,11,13,13,15,15-hexadecamethyl-	0.16
12.	31.176	5,5′-Bis [2 -(4-aminophenyl)-1H-1,3-benzimidazol]	0.18
13.	31.371	.gamma.-Tocopherol	0.27
14.	31.736	Stigmastan-3,5-diene	0.55
15.	33.778	3-Quinolinecarboxylic acid, 6,8-difluoro-4-hydroxy-, ethyl ester	0.20
16.	33.981	1H-Indole, 1-methyl-2-phenyl-	0.19
17.	34.083	Silane, trimethyl [5 -methyl-2-(1-methylethyl)phenoxy]-	0.167
18.	34.862	.beta.-Sitosterol	4.44
19.	35.159	5(1H)-Azulenone, 2,4,6,7,8,8a-hexa hydro-3,8-dimethyl-4-(1-methylethylidene)-, (8S-cis)-	0.46
20.	35.278	Benzo[h]quinoline, 2,4-dimethyl-	1.67
21.	35.650	.alpha.-Amyrin	2.62
22.	35.820	Lupeol	4.16
23.	36.260	6.beta.Bicyclo[4.3.0]nonane, 5.beta.-iodomethyl-1.beta.-isopropenyl-4.alpha.,5.alpha.-dimethyl-,	3.02
24.	36.506	Urs-12-en-24-oic acid, 3-oxo-, methyl ester, (+)-	9.80
25.	37.032	Acetic acid, 3-hydroxy-6-isopropenyl-4,8a-dimethyl-1,2,3,5,6,7,8,8a-octahydronaphthalen-2-yl ester	13.69
26.	37.150	12-Oleanen-3-yl acetate, (3.alpha.)-	21.09
27.	38.006	3-Quinolinecarboxylic acid, 6,8-difluoro-4-hydroxy-,ethyl ester	1.75
28.	38.489	A`-Neogammacer-22(29)-en-3-ol, acetate, (3.beta.,21.beta)	0.77
29.	39.048	5-Methyl-2-phenylindolizine	0.06
30.	41.302	1H-Tndol-2-carboxylic acid, 6-(4-ethoxyphenyl)-3-methyl-4-oxo-4,5,6,7-tetrahydro-, isopropyl ester	0.15

**Table 3 molecules-27-02593-t003:** Final body weight and liver index of rats in different treatment groups.

Treatment Group	Initial Body Weight (g)	Final Body Weight (g)	Percentage Increase of Body Weight (%)	Liver Index (%)
Control	227.77 ± 6.61	265.33 ± 13.63	16.49	3.08 ± 0.29
CCl_4_ (1 mL/kg bwt)	229.17 ± 7.24	237.50 ± 2.55	3.64	4.93 ± 0.37
*F. lepicarpa* (100 mg/kg bwt + CCl_4_)	228.63 ± 5.11	241.81 ± 2.93	5.76	4.15 ± 0.29
*F. lepicarpa* (200 mg/kg bwt + CCl_4_)	230.83 ± 5.75	247.77 ± 1.98	7.34	4.10 ± 0.20
*F. lepicarpa* (400 mg/kg bwt + CCl_4_)	232.58 ± 6.41	252.33 ± 3.11	8.49	4.03 ± 0.29
*F. lepicarpa* (400 mg/kg bwt) (plant control)	229.67 ± 7.20	265.75 ± 14.04	15.71	3.05 ± 0.23

All values represent the mean ± SEM of six animals (*n* = 6).

**Table 4 molecules-27-02593-t004:** Effects of *F. lepicarpa* on hepatic serum AST & ALT levels following CCl_4_ treatment.

Treatment Group	ALT Enzyme Activity (U/L)	AST Enzyme Activity (U/L)
Control	6.56 ± 0.06	11.58 ± 0.01
CCl_4_ (1 mL/kg bwt)	47.03 ± 0.33 *	42.28 ± 0.59 *
*F. lepicarpa* (100 mg/kg bwt + CCl_4_)	23.75 ± 0.41	24.20 ± 0.33
*F. lepicarpa* (200 mg/kg bwt + CCl_4_)	14.63 ± 0.37 **	19.21 ± 0.34
*F. lepicarpa* (400 mg/kg bwt + CCl_4_)	11.26 ± 0.06 **	14.41 ± 0.02 **
*F. lepicarpa* (400 mg/kg bwt) (plant control)	6.64 ± 0.07 **	11.21 ± 0.02 **

All values represent the mean ± SEM of six animals (*n* = 6). * Values differ significantly from the corresponding values of control group (*p* < 0.05). ** Values differ significantly from the corresponding values of the CCl_4_ alone treated group (*p* < 0.05).

**Table 5 molecules-27-02593-t005:** Effects of *F. lepicarpa* on hepatic GSH and MDA levels following CCl_4_ treatment.

Treatment Group	GSH (µmol/g Tissue)	MDA Formation (nmol MDA/g Tissue)
Control	9.18 ± 0.05	22.22 ± 0.01
CCl_4_ (1 mL/kg bwt)	2.33 ± 0.04 *	69.51 ± 0.12 *
*F. lepicarpa* (100 mg/kg bwt + CCl_4_)	3.87 ± 0.04	59.18 ± 0.04
*F. lepicarpa* (200 mg/kg bwt + CCl_4_)	4.03 ± 0.05 **	49.34 ± 0.07
*F. lepicarpa* (400 mg/kg bwt + CCl_4_)	5.41 ± 0.07 **	36.74 ± 0.02 **
*F. lepicarpa* (400 mg/kg bwt) (plant control)	8.32 ± 0.05 **	21.31 ± 0.02 **

All values represent the mean ± SEM of six animals (*n* = 6). * Values differ significantly from the corresponding values of control group (*p* < 0.05). ** Values differ significantly from the corresponding values of the CCl_4_ alone treated group (*p* < 0.05).

**Table 6 molecules-27-02593-t006:** Protective effects of *F. lepicarpa* on activities of hepatic antioxidant enzymes.

Treatment Group	GPx (nmol NADPH Oxidized/min/mg Protein)	GR (nmol NADPH Oxidized/min/mg Protein)	GST (nmol CDNB Conjugate Formed/min/mg Protein)	QR (nmol Dichloroindophenol Reduced/min/mg Protein)
Control	865.25 ± 0.14	215.46 ± 0.30	330.18 ± 0.04	114.06 ± 0.12
CCl_4_ (1 mL/kg bwt)	389.77 ± 0.20 *	71.71 ± 0.24 *	111.85 ± 0.04 *	59.08 ± 0.10 *
*F. lepicarpa* (100 mg/kg bwt + CCl_4_)	400.86 ± 0.11	104.86 ± 0.25	177.68 ± 0.08	80.89 ± 0.11
*F. lepicarpa* (200 mg/kg bwt + CCl_4_)	523.58 ± 0.14 **	145.89 ± 0.28 **	222.04 ± 0.04 **	100.30 ± 0.11 **
*F. lepicarpa* (400 mg/kg bwt + CCl_4_)	542.32 ± 0.06 **	182.36 ± 0.35 **	249.04 ± 0.03 **	102.04 ± 0.12 **
*F. lepicarpa* (400 mg/kg bwt) (plant control)	760.50 ± 0.06 **	203.84 ± 0.27 **	296.49 ± 0.02 **	106.38 ± 0.12 **

All values represent the mean ± SEM of six animals (*n* = 6). * Values differ significantly from the corresponding values for normal control (* *p* < 0.05). ** Values differ significantly from the corresponding values for the CCl_4_ alone treated control (** *p* < 0.05).

## Data Availability

Not applicable.

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
