# Peer review of "Suppression of Oxidative Stress and Proinflammatory Cytokines Is a Potential Therapeutic Action of Ficus lepicarpa B. (Moraceae) against Carbon Tetrachloride (CCl4)-Induced Hepatotoxicity in Rats"

_molecules, 2022, doi:10.3390/molecules27082593_

Round 1
Reviewer 1 Report
The authors present the results of their experiment on a mice model in order to test the effectivness of Ficus lepicarpa B. extract on the liver protection from oxidative stress.
The topic is interesting and innovative. The potential application of plant derivatives for the prevention or treatment of pathologies related to oxidative stress or other types of stress is attracting growing interest in the field of biomedical research.
The authors show that the administration of Ficus lepicarpa B. extract at increasing doses in mice proved to be effective, in a dose-related manner, in countering the harmful effects induced on the liver by the administration of CCl4.
The authors included in the study a wide range of biochemical markers and histological evaluations that make the data presented appear convincing.
In the opinion of this referee, the work can be accepted for publication on Molecules after addressing the points explained below.
- The first time an abbreviation is used, it is suggested to specify its meaning in full (eg. row 81: TPC, row 155 and following: GPx, GR, GST, QR).
- The meaning of "liver index" (row 108 and following) shoud be explicited, not assuming that the reader has a background of knowledge in this sense.
- It would be convenient to review the layout, bringing the tables close to the part of the text they refer to or at least on the same page
- Row 147: did the authors speculate what the specific enzyme or step might be the inhibited one? The formulation of a hypothesis in this sense could increase the value of the paper.
- Row 162: “Plant conrol group…” this sentence seems to be not consistent with the data reported in table 5. The activity of enzymes in the control is higher than in the plant supplemented group. The authors should better explain this point?
- Row 272 and following: the sentence about the weight of the rats is not clear and it could be misleading. I would suggest the authors to rephrase this sentence, better explaining the weight trend in the models according to the different administrations.
- Row 287 and following: “as a result of increased permeability of cell membranes…” For a better understanding of the text ─ even by a reader without a full background on this point ─ I would suggest the authors to better clarify this mechanism.
- Row 301-302: did the authors speculate what this mechanism might be and how GSH is restored? The formulation of a hypothesis in this sense could increase the value of the paper.
- Row 333: “The proinflammatory marker…” a verb is missing in this sentence, so its meaning is not clear
- The English form should be carefully revised. There are a number of grammar mistakes that should be corrected. In some points the English form makes difficult the comprehension of the scientific contents. Please, carefully revise the English form throughout the whole text.
- Row 549 and following: the conclusion is misleading and even seems contradictory with what was said previously in the text.
Author Response
Modifications made in revised manuscript in response to reviewers comments:
We are pleased to learn that reviewers gave positive and valuable comments to improve our manuscript. We have incorporated the comments and modifications as suggested. The details are as follows:
Reviewer #1:
- The first time an abbreviation is used has been specify its meaning in full as sugessted. Row 81: TPC, row 85: TFC, row 96: GC-MS, row 122: ALT and AST and row156: GPx, GR, GST and QR.
- The meaning of ‘liver index” (row108) has been explicited as suggested.
- The original Table 4 of GSH, MDA, AST and ALT have been separated into Table 4 (AST and ALT) and Table 5 (GSH and MDA) bringing the tables close to the part of the text they refer as suggested.
- Row 147: LPO inhibited enzymes has been added as suggested.
- Row 162: The sentence has been corrected to be consistent with the data reported in Table 6.
- Row272: The sentence has been rephrase as suggested.
- Row287: The sentence has been rephrase as suggested.
- Row 301-302: The possible mechanism of how the GSH is restored has been added as suggested.
- Row 333: The missing verb has been added as suggested.
- English language editing has been done as suggested.
- Row 549 and following: The conclusion has been revised as suggested.

Reviewer 2 Report
The topic is of interest. However, there are several concerns about the study, and needs to be improved.
1.More details are needed on how the authors selected the dose and treatment period ?
2.Why the authors did not investigate the ROS formation?
3.It is suggested to discuss about the role of Ficus lepicarpa B. (Moraceae) on mitochondria.
4.Please add your future perspective and suggestion about the possible role of antioxidant therapy..
5.It is suggested to use these papers for discussion part and bold the novelty of your study :
-Fard, J. K., et al. "Triazole rizatriptan induces liver toxicity through lysosomal/mitochondrial dysfunction." Drug Research 66.09 (2016): 470-478.
-Eftekhari, Aziz, et al. "The effects of cimetidine, N-acetylcysteine, and taurine on thioridazine metabolic activation and induction of oxidative stress in isolated rat hepatocytes." Pharmaceutical Chemistry Journal 51.11 (2018): 965-969.
-Ahmadian, Elham, et al. "In vitro and in vivo evaluation of the mechanisms of citalopram-induced hepatotoxicity." Archives of pharmacal research 40.11 (2017): 1296-1313.
Round 2
Reviewer 1 Report
The authors have satisfactorily addressed the concerns raised by the reviewers. I am pleased to recommend the manuscript for publication in Molecules in its current form.
Author Response
March 29, 2022
To,
Lien Wang
Asistant Editor
Molecules
Dear Asistant Editor,
Thank you very much for your email dated March 18, 2022 regarding our manuscript, Manuscript ID: molecules-16211894 Title “Suppression of oxidative stress and proinflammatory cytokines is a potential therapeutic action of Ficus lepicarpa B. (Moraceae) against carbon tetrachloride (CCl4)-induced hepatotoxicity in rats.” We have revised our manuscript in response to reviewer comments. We hope that revised manuscript will be acceptable for its publication to Molecules.
Looking forward to your response,
With regards,
Yours sincerely
Mohammad Iqbal, PhD.
